# Association of Self-Reported Sleep Characteristics and Hip Fracture: Observational and Mendelian Randomization Studies

**DOI:** 10.3390/healthcare11070926

**Published:** 2023-03-23

**Authors:** Yan-Fei Wang, Yu-Feng Luo, Asmi Mhalgi, Wen-Yan Ren, Long-Fei Wu

**Affiliations:** 1Center for Genetic Epidemiology and Genomics, School of Public Health, Medical College of Soochow University, Suzhou 215123, China; 2Jiangsu Key Laboratory of Preventive and Translational Medicine for Geriatric Diseases, Soochow University, Suzhou 215123, China; 3Cambridge-Suda Genomic Resource Center, Jiangsu Key Laboratory of Neuropsychiatric Diseases, Medical College of Soochow University, Suzhou 215123, China

**Keywords:** sleep, fracture, aging, longitudinal study

## Abstract

Previous observational studies on the relationship between sleep characteristics and fracture have yielded contradictory results. The goal of this study was to replicate the findings in a large longitudinal cohort and then conduct a Mendelian randomization (MR) analysis to infer the causality between sleep behaviors and fracture risk. Based on data from the China Health and Retirement Longitudinal Study (CHARLS) including 17,708 participants, we found that individuals with short sleep duration (<5 h) (OR [odds ratio] = 1.62, 95% CI: 1.07–2.44) or restless sleep (OR = 1.55, 95% CI: 1.10–2.19) have a higher risk of hip fracture. A U-shaped relationship between nighttime sleep duration and hip fracture risk (*p*-nonlinear = 0.01) was observed using restricted cubic spline regression analysis. Through joint effect analysis, we found that participants with short sleep duration (<5 h) combined with midday napping could significantly decrease hip fracture incidence. We further inferred the causal relationship between self-reported sleep behaviors and hip fracture using the MR approach. Among four sleep phenotypic parameters (sleep duration, daytime napping, chronotype, and insomnia), we found a modest causal relationship between sleep duration and fracture (OR = 0.69, 95% CI: 0.48 to 0.99, *p* = 0.04). However, no causal relationship was observed for other sleep traits. In conclusion, our findings suggest that short sleep duration has a potential detrimental effect on hip fracture. Improving sleep patterns is of significance for developing hip fracture preventive strategies in the middle-aged and the elderly populations.

## 1. Introduction

Fracture is a major health risk for the elderly population due to its high mortality and high risk of disability [1,2]. In 2000, the estimated number of new osteoporotic fractures worldwide was 9.0 million, of which 1.6 million (18.2%) were at the hip, and the number of hip fractures are expected increase to 4.5 million by 2050 [3,4]. China is experiencing rapid population aging and facing great challenges from an increasing number of hip fractures [5]. As a result, identifying risk factors for hip fracture is critical for developing comprehensive preventive strategies and lowering the societal cost burden.

Hip fractures in the elderly result from multiple etiological factors, such as falls, physical activity, poor bone health, cognitive impairment, and lifestyle factors [6,7,8]. Extensive studies have shown that reduced bone mineral density (BMD) is the key determinant of the risk of hip fractures. Falls are another direct cause of hip fractures in the elderly. Some other modifiable lifestyle factors, such as external body padding, also offer great promise in the prevention of hip fracture [9]. Disruptions in the physiology of sleep and circadian rhythmicity may have an effect in bone health and increase the risk of fracture [10,11,12]. From experimental studies, mice lacking molecular clock genes (e.g., *Per* and *Cry*) presented abnormal skeletal phenotypes [13,14,15,16], implying that bone remodeling is subject to circadian regulation. Similarly, the expression of clock genes in human bone has endogenous circadian rhythmicity [17]. Sleep disturbance has also been linked to incidental fractures in epidemiologic studies. In the Nurses’ Health Study, for example, postmenopausal women who had had rotating night shifts work for a long time had an increased risk of hip and wrist fractures compared to women who never worked the night shift [18]. Interestingly, individuals with sleep disorders are more likely to consume caffeine or medications (e.g., benzodiazepines), both of which further increase the risk of falls and fractures [19,20]. Other sleep behaviors, such as sleep duration, daytime napping, chronotype, sleep quality, insomnia, and snoring, have also been linked to bone mineral density (BMD) and fracture risk [11,21,22,23]. Even though some pathophysiological research has suggested that genetics, brain structure, cytokines, hormones, and increased food intake may play a role in the link between sleep disorder and hip fracture, the underlying mechanism is not fully understood [24,25,26,27].

Observational studies have suggested associations between sleep behaviors and fracture risk, but the findings have been inconsistent. For sleep duration, data from the Women’s Health Initiative revealed that self-reported short sleep was associated with lower BMD and a higher risk of fracture [28]. Conversely, the Study of Osteoporotic Fractures (SOF) found that long sleep duration was associated with an increased risk of non-spine fractures in older postmenopausal women [29]. Several studies recently suggested a U-shaped relationship between sleep duration and fracture risk [30,31]. Surprisingly, the role of sleep traits on bone health varies by fracture site and gender [32,33]. For example, poorer sleep, such as shorter sleep duration, sleepiness, and snoring, were associated with vertebral fracture but not hip fracture in another longitudinal cohort study [33]. Such controversial findings may be in part due to differences in the populations studied. In addition, it is also important to consider some other unmeasured variables (for example, outdoor activity, morbidities, and depression), as well as the joint effects of sleep traits. Of note, such epidemiologic associations are vulnerable to reverse causation biases because fracture can influence sleep duration and quality. As a result, current observational studies have a limited ability to further disentangle the potential causal relationship between sleep characteristics and fracture.

Mendelian randomization (MR) investigates causal effects of exposure (sleep behaviors) and outcome (fracture) using genetic variants as instrumental variables [34]. Because the genetic alleles are randomly assigned, this method is less susceptible to unmeasured confounding [35]. Recent reports have identified several genetic variants that are robustly associated with sleep traits (e.g., self-reported insomnia, chronotype, napping, and sleep duration) through genome-wide association studies (GWAS) [36,37,38,39]. The public availability of genetic data allows us to investigate causal relationships between sleep traits and fracture in MR studies.

Previous observational studies have discovered that sleep habits influence the risk of fracture. However, it is unclear whether there is a nonlinear relationship between sleep duration and fracture. The frequency of daytime naps increases with age; is there an association between this and hip fracture, particularly in the elderly? Furthermore, what are the causal roles of self-reported daytime napping, chronotype, and other sleep phenotypes in fracture? With this in mind, we carried out an observational study to examine the effects of nighttime sleep duration, midday napping, and sleep quality on hip fracture within a large Chinese middle-aged and older cohort. Furthermore, we used genetic variants that are associated with sleep traits (sleep duration, chronotype, daytime napping, and insomnia) and fracture to infer their possible causal relationships in a systematic MR study.

## 2. Materials and Methods

### 2.1. Participants

This study used the China Health and Retirement Longitudinal Study (CHARLS) dataset to investigate socioeconomic determinants and consequences of aging. This is an on-going, nation-wide representative longitudinal health survey of residents aged over 45 years and older from 450 villages or communities in 150 districts of 28 participating provinces, with follow-up every two years. The description and questionnaire of the CHARLS have been described elsewhere [40,41]. In this study, we utilized data from the CHARLS 2011 (wave 1) and 2015 (wave 3). The baseline survey of wave 1 included a total of 17,708 participants, and wave 3 included 20,517 participants. After consolidating the baseline files, 11,912 participants remained. Besides, 14 subjects were excluded due to lack of follow-up in 2015. The 195 subjects that reported hip fracture at baseline and the 236 subjects missing sleep duration and midday napping information in the baseline survey were excluded from the longitudinal analysis. A total of 495 participants aged ≤ 45 and 43 participants with BMI ≤ 10 or ≥50 were excluded. Finally, a total of 10,929 adults were included in the following analysis (Figure 1). This study was performed in line with the principles of the *Declaration of Helsinki*. This study was approved by the Biomedical Ethics Review Committee of Peking University (IRB00001052-11015). All subjects enrolled signed informed consent.

### 2.2. Assessment of Sleep Habits, Hip Fracture, and Potential Confounders

Information on sleep duration and daytime nap duration was recorded using a self-reported questionnaire. Napping duration was collected by asking: “During the past month, for how long did you take a nap after lunch on average?” Midday napping was divided into no napping (0 min) or 1–30, 31–60, 61–90, or >90 min. Sleep duration was assessed by the following question: “During the past month, how many hours of actual sleep did you get at night (average hours for one night)?” Sleep duration was categorized into 5 groups: <5, 5–6, 6–7, 7–8, 8–9, or >9 h. Sleep quality was obtained by asking: “My sleep was restless during the last week” with the following responses: rarely or none of the time (<1 day), some or a little of the time (1–2 days), occasionally or a moderate amount of the time (3–4 days), or most or all the time (5–7 days). Sleep quality was also evaluated by asking: “In the last month, how much difficulty did you have with sleeping, such as having trouble falling asleep, waking up frequently during the night, or waking up too early in the morning?” and the response was categorized: none, mild, moderate, severe, and extreme. Hip fracture data was collected using a self-reported questionnaire which asked: “Have you been diagnosed with hip fracture by a doctor?” If the respondents answered “yes”, then they were defined as hip fracture patients. Basic characteristics of each participant were collected including age, gender, body mass index (BMI), smoking status, drinking status, personal medical histories, and history of chronic diseases, such as hypertension, dyslipidemia, and high blood sugar/diabetes. BMI was calculated as weight in kilograms divided by height in meters squared.

### 2.3. Two-Sample MR Analysis

The data of sleep-related phenotypes were obtained from the IEU OpenGWAS database (URL: https://gwas.mrcieu.ac.uk/; accessed on 19 December 2022), a database of 245,322,865,636 genetic associations from 42,335 GWAS summary datasets, for querying or download. The sleep-related phenotypes used in this study were sleep duration, daytime napping, chronotype, and insomnia.

The GWAS summary data for hip fracture was obtained from the GEnetic Factors for OSteoporosis Consortium (GEFOS, URL: http://www.gefos.org/; accessed on 19 December 2022), which included 25 cohorts (37,857 cases and 227,116 controls). These cohorts were predominantly of European descent (n = 15) or from North America (n = 8), Australia (n = 1), and east Asia (n = 1).

To establish a causal effect of sleep-related traits on hip fracture, the MR method needs to satisfy the following three assumptions: (i) the genetic variant (Instrumental variable) is robustly associated with hip fracture (Exposure); (ii) the genetic variant does not share common causes (potential confounding factors) with hip fracture (Outcome); and (iii) the genetic variant affects hip fracture (Outcome) exclusively through its effect on sleep-related traits (Exposure).

Five main methods are used to estimate causal effects: the random effects inverse-variance weighted (IVW) method was utilized in the main MR analyses. And MR-Egger regression, the weighted median, simple mode, and weighted mode were performed as complementary analyses. The different methods can provide valid evidence under different conditions. We utilized the IVW method as the primary analysis for its efficiency to estimate the causal effect. The weighted median was used as an auxiliary method when the heterogeneity was significant, and the MR-Egger regression method was used to assess the pleiotropy by intercept test. Several sensitivity analyses were used to check and correct for the causal estimates. Heterogeneity between SNPs included in each analysis was first tested using the Cochran Q test, and if heterogeneity existed, then random effects IVW was used. A combined sensitivity analysis was then performed to verify the robustness of our results. The intercept of the MR-egger method was used to test for horizontal multiplicity, and the MR multiplicity residual sum and outlier method (MR-PRESSO) was used to detect potential outliers [42]. A leave-one-out analysis was performed to evaluate the stability of these genetic variants on sleep-related traits.

### 2.4. Statistical Analyses

Baseline data was presented as mean ± standard deviation for continuous variables and number (percentage) for categorical variables. The differences in baseline characteristics between groups were compared using χ^2^ analysis for categorical variables and analysis of variance or Mann-Whitney U tests for continuous variables. Logistic regression was applied to analyze the association of sleep duration and midday napping with hip fracture. The results are presented as multivariable adjusted odds ratio (OR) and 95% confidence interval (CI). Potential covariates included in the multivariable adjusted model were age, gender, smoke, drink status, BMI, hypertension, dyslipidemia, diabetes, or high blood sugar. For the joint analysis, a total of 30 categories were defined by combining six categories of sleep duration with five categories of nap duration. Then, a logistic regression was carried out on each of the 30 categories and the outcome to obtain the relationship with the outcome. We assessed potential nonlinear trends of incident hip fracture risk by restricted cubic spline regression. Statistical analyses were conducted using SAS 9.2 (SAS Institute, Cary, NC, USA) or the R package “TwoSampleMR” and “MRPRESSO”, and *p* < 0.05 was considered statistically significant.

## 3. Results

### 3.1. Observational Analysis

Table 1 shows the demographic characteristics of participants by nighttime sleep and midday napping duration. A total of 10,929 participants who were in the inception cohort in 2011 were included in this study. Among them, 888 (8.1%) participants reported sleeping >9 h/night and 1249 (11.4%) reported midday napping for >90 min. The average length of nighttime sleep duration and midday napping time were 6.34 ± 1.88 h and 32.60 ± 42.90 min, respectively. Individuals who reported sleep duration >9 h/night or midday napping >90 min tended to be younger, male, smokers, and drinkers, compared to those who reported sleep duration less than 5 h/night or no napping behavior. Participants who reported >9 h/night of sleep were less likely to have history of diabetes or high blood sugar, whereas participants who reported midday napping >90 min had increased prevalence of hypertension, diabetes or high blood sugar, and hyperlipidemia compared with the reference groups.

During 4 years of follow-up, a total of 224 incidental hip fracture cases were observed. Compared with participants sleeping for 6–7 h/night, the ORs (95% CIs) of hip fracture were 1.62 (1.07–2.44) for <5 h/night, 0.80 (0.47–1.36) for 5–6 h/night, 0.88 (0.56–1.39) for 7–8 h/night, 1.21 (0.80–1.82) for 8–9 h/night, and 0.99 (0.55–1.77) for >9 h/night, respectively. After adjusting for covariates, the results remain consistent (Table 2). Restricted cubic spline regression analysis was further used to model the association between nighttime sleep duration and hip fracture. An approximately U-shaped association between sleep duration and hip fracture was observed in middle-aged and older Chinese adults (*p*-nonlinear = 0.01), indicating that elderly persons with both insufficient and excessive sleep duration periods at night had a higher risk of hip fracture (Figure 2).

Furthermore, we examined the association of midday napping with hip fracture risk. Compared with participants who napped for 31–60 min, the multivariable adjusted ORs (95% CIs) of hip fracture were 1.27 (0.87–1.85) for non-nappers, 1.46 (0.94–2.27) for those who reported 0–30 min, 0.56 (0.17–1.82) for 61–90 min, and 1.49 (0.92–2.41) for midday napping for >90 min. This suggested that baseline midday napping time had no effect on the risk of hip fracture. Besides, upon conducting a stratified analysis using selected covariates (e.g., gender, BMI, and disease status), the association of hip fracture with sleep duration or midday napping duration seemed to be stronger among individuals who reported having diabetes or high blood sugar (Figure 3 and Figure 4). Furthermore, we assessed the joint effect of nighttime sleep duration and midday napping time on fracture risk. Compared with participants with short sleep duration (<5 h) and no midday napping behavior, we found that moderately prolonged napping time (0–30 min: OR = 0.438, 95% CI: 0.25–0.776; and >90 min: 0.455, 95% CI: 0.21–0.989) significantly decreased the risk of hip fracture (Figure 5).

Additionally, sleep quality was assessed by the number of days with restless sleep in the past week (<1, 1–2, 3–4, and 5–7 days/week), and we found that participants who suffered severe restless sleep (5–7 days/week) had a significantly increased risk of hip fracture (OR 1.55, 95% CI: 1.10–2.19) compared to those who rarely or never experienced restless sleep (Table 3). However, there is no statistical association between difficulty falling asleep or waking up and hip fracture.

### 3.2. Two-Sample MR Analysis

Among four sleep phenotypic studies we conducted (e.g., sleep duration, daytime napping, chronotype, and insomnia), we found a modest causal relationship between sleep duration and fracture in the main inverse-variance weighted analyses (OR = 0.69, 95% CI: 0.48 to 0.99, *p* = 0.04) (Figure 6). The scatter plot and funnel plot are shown in Appendix A. For instrumental variable (IV) selection, a total of 71 single-nucleotide polymorphisms (SNPs) that have a robust association with sleep duration at the threshold of statistical significance (*p* < 5 × 10^−8^) were selected. To remove bias from linkage disequilibrium (LD), a clumping process was conducted with the European population and LD between SNPs (R^2^ < 0.1, kb = 3000). Finally, one SNP was excluded due to LD, leaving 70 SNPs as IVs for further analysis (Appendix A). Because the instrumental variable was selected only from the European population, 41 SNPs were not found in the GEFOS data sets. In addition, we removed three SNPs with intermediate allele frequency palindromes. As a result, estimates for 26 SNPs were included in the analysis of sleep duration and hip fracture. For sensitivity analysis, Cochran’s Q test showed that there is no significant heterogeneity for the causal effect of sleep duration on hip fracture (Q_pval > 0.05). The MR-Egger intercept test demonstrated that our results were not influenced by pleiotropy (*p* = 0.41). MR-PRESSO analysis showed no horizontal pleiotropy (*p*_mr-Presso_ = 0.183). Lastly, the leave-one-out sensitivity analysis confirmed the stability of the causal inference (Appendix A). However, we failed to observe a causal relationship of daytime napping, chronotype, and insomnia with fracture.

## 4. Discussion

In this study, we replicated previous observational findings by examining the association of self-reported sleep duration, midday napping, and sleep quality with hip fracture in a large Chinese middle-aged and elderly cohort [30,43]. Our results suggested that participants who self-reported short sleep duration and poor sleep quality (e.g., restless sleep) were associated with higher risk of fracture. Moreover, a U-shaped relationship was observed between sleep duration and fracture risk. Furthermore, we found a joint effect of sleep duration and midday napping on the risk of hip fracture. In the two-sample MR analysis, the results suggested that genetically determined self-reported sleep duration was potentially negatively associated with the risk of hip fracture. However, no evidence of a causal relationship between other sleep traits, such as chronotype, daytime napping, and insomnia, and fracture was found in the MR analysis.

Of the sleep characteristics, previous observational studies investigated the association of self-reported sleep duration with fracture but reported conflicting results. In the Study of Osteoporotic Fractures, postmenopausal women with daily napping and longer sleep duration (>10 h) were significantly associated with greater risk of hip fracture [29]. In women aged 50 to 79 years (n = 157,206) in the Women’s Health Initiative, short sleep duration (≤5 h/night) was associated with increased risk of all fractures [28]. They also raised the intriguing point that long sleep was associated with an increased risk of recurrent falls, but it had no effect on fracture despite most fractures occurring as a result of a fall. Another longitudinal Nurses’ Health Study (n = 122,254) with over 12–14 years of follow-up found a moderate inverse association between sleep duration and vertebral fracture risk, implying that short sleep duration (≤5 h/night) is detrimental to bone health [33]. In the current study, we found that participants (n = 17,708) who slept less than 5 h per night had a higher risk of hip fracture (OR = 1.62, 95% CI: 1.07–2.44) in a Chinese national longitudinal study, but there was no association between fracture risk and longer duration of sleep (OR = 0.99, 95% CI: 0.55–1.78). Consistent with a recent study in a European population [30], we confirmed the U-shaped relationship between sleep duration and hip fracture (*p*-nonlinear = 0.01) in a Chinese population. The underlying mechanism of such nonlinear associations may be mediated by genetics and brain structure in the middle-aged and older population [24].

Besides sleep duration, there have been fewer reports on the association of other sleep patterns (e.g., midday napping, insomnia, and sleep quality) with incidental hip fracture, especially in the Chinese population. We found no association between midday napping and hip fracture in this study, but poor sleep quality, as defined by restless sleep, showed increased odds of hip fracture after adjusting for multiple covariates (OR: 1.55, 95% CI: 1.10–2.19). Contrary to a previous study [32], we did not observe a gender difference between sleep duration and risk of hip fracture in this middle-aged and elderly Chinese population.

Prior studies have investigated sleep behaviors individually with fracture. Emerging evidence highlights the importance of multiple dimensions of sleep in the development of fractures given that sleep behaviors may affect each other [44]. For example, a study investigated the association between poor sleep, as defined by four sleep-related traits (sleep duration, sleep difficulty, snoring, and sleepiness), with fracture risk [33]. Another study developed a sleep risk score to assess the overall relationship between the combination of four sleep behaviors and fracture risk [30]. Given that short nighttime sleep duration may increase midday napping time, we carried out a joint effects analysis of sleep duration and midday napping on hip fracture incidence. Our results showed that short sleep duration (<5 h/night) combined with midday napping time was associated with decreased risk of hip fracture. As a result, it is critical to begin preventive care for elderly people with shorter sleep duration by increasing napping time or frequency as a complement.

Sleep is a complex phenotype that involves numerous physiological and pathological processes. The mechanisms underlying the association of sleep traits with incidental fractures are still unknown. Previous studies have suggested that impaired cognitive function or depression, falls, decreased muscle strength, bone mineral density (BMD), and comorbidities may mediate this association [11,31,33,45,46]. For example, short/long sleep duration was frequently comorbid with falls, osteoporosis, pain, mental disorders, and obstructive sleep apnea [45,47]. Some biological research also suggested that the circadian rhythm was involved in bone remodeling. For instance, sleep disorder was relevant to the disrupted circadian rhythm [48], which also influences fractures via inflammation factors, metabolites, melatonin, and other hormonal factors [25,26,49].

MR analyses apply genetic variants as instrumental variables to assess the causality of exposure–outcome associations, which are less susceptible to residual confounders and reverse causation [34]. In this study, we used the MR approach to systematically detect the causal relationship of four sleep traits with fracture risk. Using genetic variants associated with sleep phenotypes as instrument variables, a weak genetic association between sleep duration and fracture risk was observed. These results provide us with some causal clues in terms of genetics between sleep duration and fracture, which is more of a complement to the results of observational studies. This information can be used to inform people who are at greater risk of hip fracture in advance and to take more effective protective measures to reduce the incidence of hip fracture. Unexpectedly, we found no causal relationship between self-reported insomnia, chronotype, and daytime napping with fracture risk, which contradicts some observational studies. Such inconsistency is probably due to residual confounders that were not controlled for in observational studies, such as history of falls or OSA. Furthermore, the null causal association in an MR study is likely to be a consequence of limited statistical power due to the low phenotypic variance explained by the genetic instruments.

Although we conducted comprehensive methodologies to dissect the relationship between sleep characteristics and fracture, there are still several limitations. First, information on sleep behaviors in both observational and MR studies was obtained through a self-reported questionnaire, which may lead to misclassification of exposures and outcomes. Further studies with more objective measurements of sleep phenotypes are needed to better understand their relationships. Second, although we adjusted for several potential confounders that might affect the relationship between sleep patterns and hip fracture in our observational study, the results should be interpreted with caution given that other confounders, such as obstructive sleep apnea, were not excluded. Therefore, we were not yet able to draw conclusions regarding the effect of some self-reported sleep traits on the risk of hip fracture. Third, the demographic makeup in our observational study was exclusively middle-aged and older Chinese adults, which may not fully extrapolate to other populations of all ages or other ethnicities. Fourthly, in order to verify the reliability of the study, we further calculated the test efficiency of the correlation study. Despite the seemingly large sample size we used, the incidence of hip fracture (about 2%, 224/10,929) is low and can be described as a rare event. According to the incidence of hip fracture (2%), the significance level (*p* = 0.05), and the incidence difference between groups (0.5%), we used a sample size (n = 10,929) that could achieve a power of 0.70. Thus, when investigating a rare event, a large sample size may be required to achieve adequate power, even though the outcome is rare.

## 5. Conclusions

In conclusion, our observational study supported an association of short sleep duration and poor sleep quality with increased risk of fracture. Furthermore, we found that combining midday napping time with short sleep duration (<5 h) could reduce the risk of hip fracture. According to the MR analysis, self-reported sleep duration appeared to be a potential risk factor for hip fracture. Our study suggests that middle-aged and elderly people should keep adequate sleep, which may have protective implications for preventing hip fractures. It also provides evidence to better understand the relationship between sleep characteristics and hip fractures, which has public health implications. It also provides valuable information for preventing fractures in middle-aged and elderly people.

## Figures and Tables

**Figure 1 healthcare-11-00926-f001:**
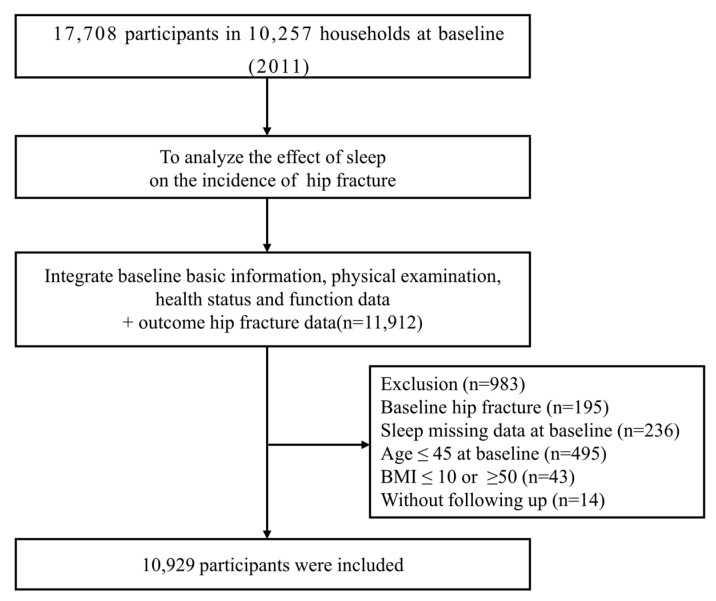
Flowchart of baseline data exclusions.

**Figure 2 healthcare-11-00926-f002:**
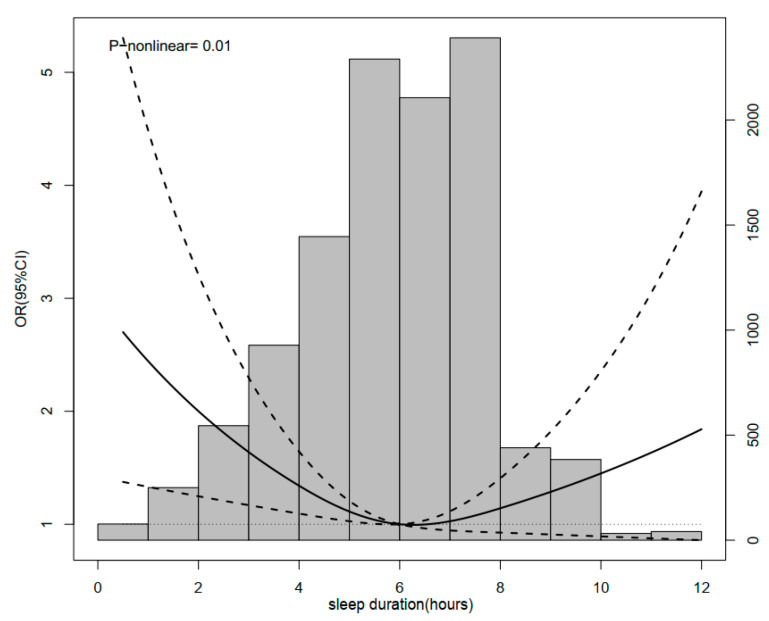
Restrictive cubic spline diagram between sleep duration and hip fracture.

**Figure 3 healthcare-11-00926-f003:**
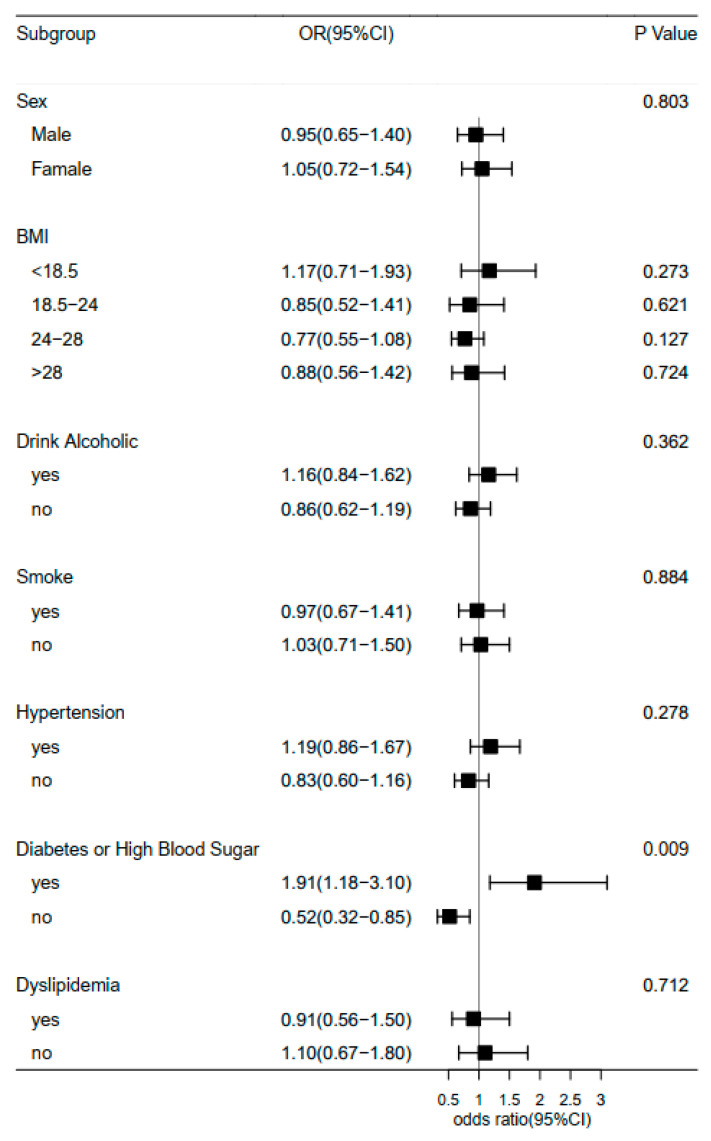
Forest plot of logistic regression analysis of sleep duration and risk of hip fracture, stratified by baseline information.

**Figure 4 healthcare-11-00926-f004:**
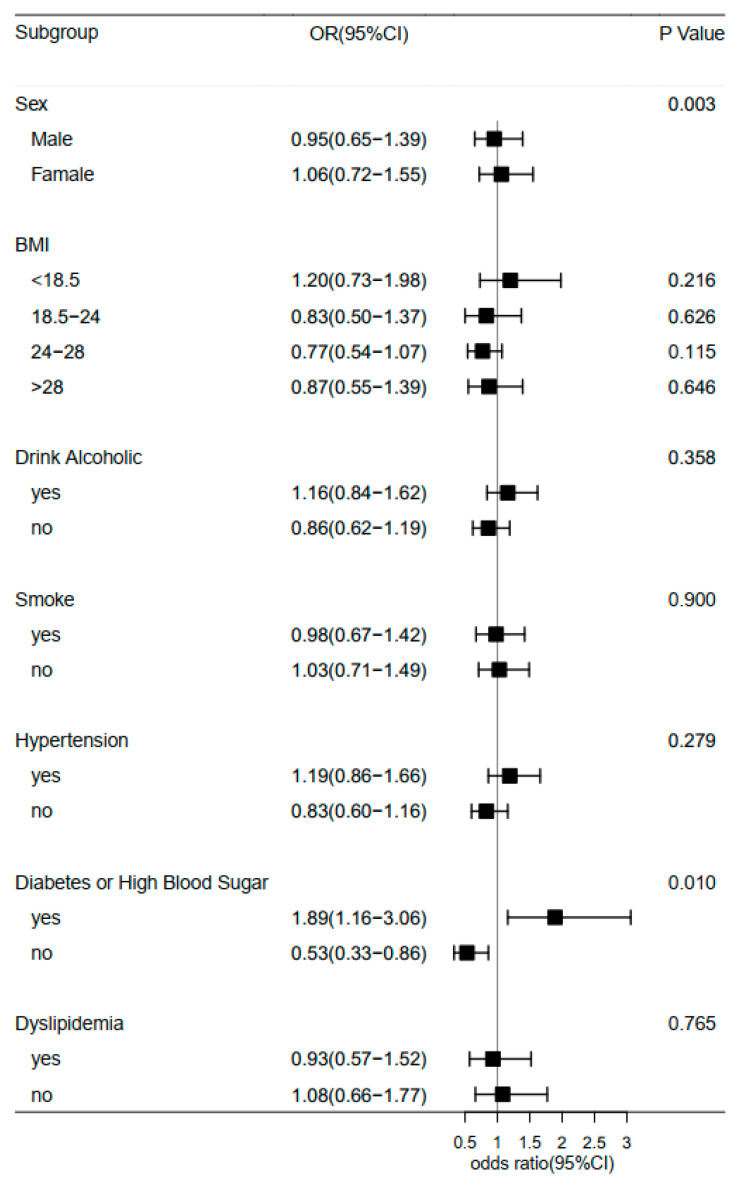
Forest plot of logistic regression analysis of midday napping and risk of hip fracture stratified by baseline information.

**Figure 5 healthcare-11-00926-f005:**
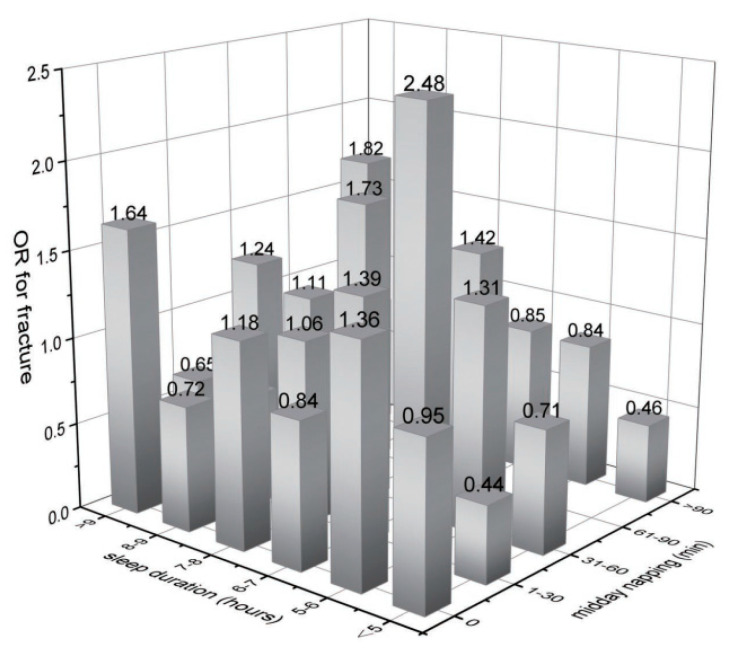
Joint effects of sleep duration and midday napping on hip fracture.

**Figure 6 healthcare-11-00926-f006:**
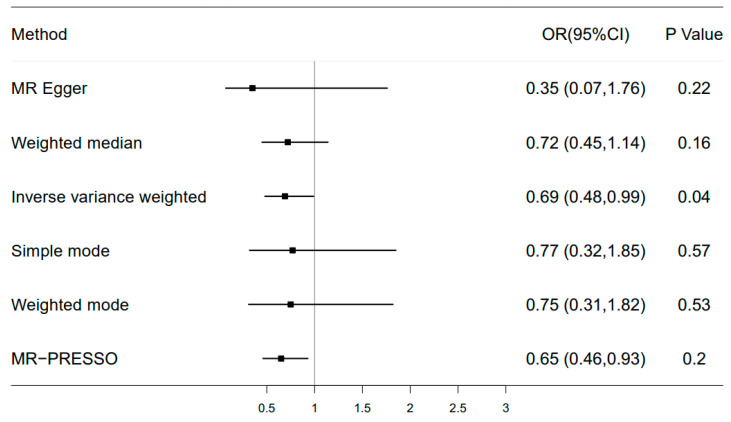
Forest plot of Mendelian randomization for the relationship between sleep duration and hip fracture risk. The random effects inverse-variance weighted (IVW) method was utilized in the main MR analyses. MR-Egger regression, the weighted median, simple mode, and weighted mode were performed as complementary analyses.

**Table 1 healthcare-11-00926-t001:** Baseline characteristics of the study participants according to sleep duration and midday napping (n = 10,919).

Variables	Sleep Duration, Hours/Night							Midday Napping, Minutes					
	<5 (n = 1821)	5–6 (n = 1441)	6–7 (n = 2305)	7–8 (n = 2117)	8–9 (n = 2357)	>9 (n = 888)	*p*-Value ^a^	0 (n = 5123)	1–30 (n = 1826)	31–60 (n = 2399)	61–90 (n = 332)	>90 (n = 1249)	*p*-Value ^a^
Age (mean ± SD)	61.65 ± 9.37	59.93 ± 8.70	58.92 ± 8.84	57.76 ± 8.37	58.54 ± 9.02	59.79 ± 9.68	<0.0001	58.93 ± 8.83	59.49 ± 8.88	59.39 ± 9.25	59.41 ± 8.70	60.11 ± 9.53	0.0006
Male (n,%)	699 (38.4)	666 (46.2)	1146 (49.7)	1024 (48.4)	1204 (51.1)	430 (48.4)	<0.0001	2070 (40.4)	847 (46.4)	1336 (55.7)	204 (61.4)	712 (57.0)	<0.0001
Hypertension (n,%)	462 (25.4)	354 (24.6)	526 (22.8)	494 (23.3)	524 (22.2)	206 (23.2)	0.1869	1038 (20.3)	494 (27.1)	615 (25.6)	85 (25.6)	334 (26.7)	<0.0001
Dyslipidemia (n,%)	161 (8.8)	142 (9.9)	211 (9.2)	173 (8.2)	206 (8.7)	65 (7.3)	0.2989	340 (6.6)	207 (11.3)	251 (10.5)	42 (12.7)	118 (9.4)	<0.0001
Diabetes or High Blood Sugar (n,%)	103 (5.6)	94 (6.5)	122 (5.3)	136 (6.4)	103 (4.4)	41 (4.6)	0.0138	213 (4.2)	136 (7.4)	163 (6.8)	21 (6.3)	66 (5.3)	<0.0001
Smoke (n,%)	642 (35.3)	561 (38.9)	945 (41.0)	882 (41.7)	958 (40.6)	346 (39.0)	0.0006	1817 (35.5)	673 (36.9)	1068 (44.5)	178 (53.6)	598 (47.9)	<0.0001
Drinking alcoholic beverages (n,%)	510 (28.0)	472 (32.8)	804 (34.9)	729 (34.4)	799 (33.9)	269 (30.3)	<0.0001	1463 (28.6)	599 (32.8)	894 (37.3)	143 (43.1)	484 (38.8)	<0.0001
BMI (n,%)	-						0.5812	-	-	-	-	-	0.5154
<18.5	177 (9.7)	94 (6.5)	141 (6.1)	104 (4.9)	121 (5.1)	58 (6.5)	-	371 (7.2)	108 (5.9)	137 (5.7)	18 (5.4)	61 (4.9)	-
18.5–24	989 (54.3)	776 (53.9)	1172 (50.8)	1060 (50.1)	1238 (52.5)	494 (55.6)		2823 (55.1)	913 (50.0)	1211 (50.5)	150 (45.2)	632 (50.6)	-
24–28	438 (24.1)	404 (28.0)	683 (29.6)	661 (31.2)	695 (29.5)	228 (25.7)		1333 (26.0)	546 (29.9)	734 (30.6)	124 (37.3)	372 (29.8)	-
>28	164 (9.0)	144 (10.0)	282 (12.2)	270 (12.8)	272 (11.5)	91 (10.2)		512 (10.0)	222 (12.2)	289 (12.0)	37 (11.1)	163 (13.1)	-

Abbreviation: BMI = body mass index. ^a^
*p*-values were derived from analysis of variance or Mann-Whitney U tests for continuous variables according to data distribution and χ^2^ tests for category variables.

**Table 2 healthcare-11-00926-t002:** Association of sleep duration and midday napping with hip fracture.

Variable	Cases/Total	Not Adjusted OR (95% CI)	Adjusted OR (95% CI) ^a^
Sleep duration, hours/night			
<5.0	55/1821	**1.62 (1.07–2.44)**	**1.52 (1.00–2.32)**
5.0–6.0	21/1441	0.80 (0.47–1.36)	0.79 (0.46–1.34)
6.0–7.0	43/2305	1.00 (ref)	1.00 (ref)
7.0–8.0	34/2117	0.88 (0.56–1.39)	0.92 (0.58–1.45)
8.0–9.0	53/2357	1.21 (0.80–1.82)	1.26 (0.84–1.91)
>9.0	18/888	0.99 (0.55–1.77)	0.99 (0.55–1.78)
Midday napping, minutes			
0	105/5123	1.22 (0.84–1.78)	1.27 (0.87–1.85)
1–30	45/1826	1.50 (0.97–2.32)	1.46 (0.94–2.27)
31–60	40/2399	1.00 (ref)	1.00 (ref)
61–90	3/332	0.55 (0.17–1.78)	0.56 (0.17–1.82)
>90	31/1249	1.49 (0.92–2.42)	1.49 (0.92–2.41)

Abbreviations: CI = confidence interval; OR = odds ratio. ^a^ Adjusted for age (continuous), sex, body mass index (<18.5, 18.5–24, 24–28, >28), smoke (yes or no), drinking alcoholic beverages (yes or no), hyperlipidemia (yes or no), dyslipidemia (yes or no), and diabetes or high blood sugar (yes or no). Remarks: Values in bold in the table are meaningful results; ref: indicates the reference group.

**Table 3 healthcare-11-00926-t003:** Association of sleep quality with hip fracture.

Variable	Cases/Total	Not Adjusted OR (95% CI)	Adjusted OR (95% CI) ^a^
My sleep was restless, days/week			
<1	91/5328	1.00 (ref)	1.00 (ref)
1–2	33/1777	1.10 (0.73–1.65)	1.09 (0.72–1.64)
3–4	42/1634	**1.49 (1.02–2.17)**	1.44 (0.98–2.11)
5–7	58/2130	**1.62 (1.16–2.28)**	**1.55 (1.10–2.19)**
Difficulty falling asleep or waking up			
None	26/1471	1.00 (ref)	1.00 (ref)
Mild	10/413	1.50 (0.71–3.16)	1.59 (0.75–3.38)
Moderate	7/338	1.27 (0.54–2.97)	1.34 (0.57–3.16)
Severe	6/207	1.80 (0.72–4.45)	1.83 (0.73–4.60)
Extreme	0/47	-	-

Abbreviations: CI = confidence interval; OR = odds ratio. ^a^ Adjusted for age (continuous), sex, body mass index (<18.5, 18.5–24, 24–28, >28), smoke (yes or no), drinking alcoholic beverages (yes or no), hyperlipidemia (yes or no), dyslipidemia (yes or no), and diabetes or high blood sugar (yes or no). Remarks: Values in bold in the table are meaningful results; ref: indicates the reference group.

## Data Availability

Data are available and can be downloaded from http://charls.pku.edu.cn/, accessed on 11 February 2023.

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
