# Peer review of "Association of Self-Reported Sleep Characteristics and Hip Fracture: Observational and Mendelian Randomization Studies"

_healthcare, 2023, doi:10.3390/healthcare11070926_

Round 1

Reviewer 1 Report

In their study titled: “Association of Self-reported Sleep Characteristics and Hip Fracture: Observational and Mendelian Randomization Studies” the authors present a sophisticated statistical analysis of survey data to address the potential causal relationship between sleep issues and hip fractures. The statistical methodology is extensive and seems to be thorough. My main concerns are regarding the clarity of the reporting of the methods and results.

Major concerns

Hip fractures result from falls and poor bone health (and other factors such as tripping hazard, body padding etc. also play a role). Please mention this multifactorial etiology in the introduction of the manuscript as this is important in understanding how sleep may influence hip fracture incidence.

Testing the association between sleep traits and hip fractures by using genetic variants as instrumental variable would (if I’ve understood correctly) potentially reveal unbiased and presumably causal relationships. However, this would not be a modifiable risk factor for hip fracture (prevention), as these sleep issues are genetic, not lifestyle related. Please discuss this issue.

As described in the manuscript, obstructive sleep apnoea (OSA) is a potential confounder as it can lead to poor sleep as well as falls (which may result in hip fracture). Pain related to metastasized cancer is another potential confounder as it can disturb sleep and also affect bone health (in various ways). Neurological conditions that are associated with pain could also affect sleep as well as the likelihood of falls. Please mention confounders other than OSA.

Please provide power calculations for this study. Although the sample size is seemingly large, the outcome is relatively rare.

The results as they are currently presented are somewhat confusing. Which results are based on instrumental variable modelling and which are not? Please also give a clear, structured overview of how the various analyses produced various (sometimes seemingly contradictory) results, and please provide context and interpretation for these findings.

Minor concerns

-  Abstract: “…genetically predicted self-reported sleep duration …” I can understand what is meant by this after reading the full manuscript, but on first reading the abstract only, I was confused. Can you please rephrase so that the abstract can be read as stand-alone.

-  Introduction, lines 33-34: please list the time and place for this statistic.

-  In the methods section, please provide more information on each of the surveys that were used (three surveys; one baseline and two follow-ups, at two-year intervals each?) and the participation numbers for each.

-  After-lunch napping is mentioned specifically (methods, line 112). What about late-morning or early evening naps?

-  Did you test for an interaction effect between sleep duration and napping? If so, can you describe in the methods exactly how this was done?

-  If this study is modelled on previous studies that have the utilised genetic variants as instrumental variables in epidemiological, observational research, please refer to these.

-  Methods page 5 line 186 “… increased risk of hypertension”. Please use ‘prevalence’ instead of ‘risk’.

-  The U curve is shown in Figure 1 but I can’t see the ‘U’ pattern in the results presented in Table 2, in the adjusted or unadjusted results. Please provide a clear explanation for this.

-  Figs 3 & 4: in the figure legends, please explain what modelling was applied to produce these results.

-  At the end of the manuscript (Page 8, line 369) there is a statement: “This is an observational study with no ethical approval”. This is at odds with Page 3 Lin 104-105: “This study was performed in line with the principles of the Declaration 104 of Helsinki. This study was approved by Biomedical Ethics Review Committee of Peking 105 University. All subjects enrolled signed informed consent.” I presume the latter is the correct statement?

Author Response

Dear Reviewer,

Response to Reviewer 1 Comments

In their study titled: “Association of Self-reported Sleep Characteristics and Hip Fracture: Observational and Mendelian Randomization Studies” the authors present a sophisticated statistical analysis of survey data to address the potential causal relationship between sleep issues and hip fractures. The statistical methodology is extensive and seems to be thorough. My main concerns are regarding the clarity of the reporting of the methods and results.

Major concerns

Point 1. Hip fractures result from falls and poor bone health (and other factors such as tripping hazard, body padding etc. also play a role). Please mention this multifactorial etiology in the introduction of the manuscript as this is important in understanding how sleep may influence hip fracture incidence.

Response 1: Thank the Reviewer for this good suggestion. Hip fractures in the elderly result from multiple etiological factors, such as falls, physical activity, poor bone health, cognitive impairment, and lifestyle factors[1-3]. Extensive studies have shown that reduced bone mineral density (BMD) is the key determinant of the risk of hip fractures. Fall is another direct cause of hip fractures in the elderly. Some other modifiable lifestyle factors, such as external body padding, also offer great promise in the prevention of hip fracture[4]. To better understand the effects of sleep on hip fracture incidence, we have included the above background in the Introduction section (Please refer lines 40-45, Page 1).

Point 2. Testing the association between sleep traits and hip fractures by using genetic variants as instrumental variable would (if I’ve understood correctly) potentially reveal unbiased and presumably causal relationships. However, this would not be a modifiable risk factor for hip fracture (prevention), as these sleep issues are genetic, not lifestyle related. Please discuss this issue.

Response 2: We thank the reviewer for raising this important issue. The Mendelian Randomization study (MR) uses genetic variation as an instrumental variable to infer whether risk factors have a causal effect on health outcomes[5]. Its results provide us with some causal clues in terms of genetics between sleep duration and fracture, which is more of a complement to the results of observational studies. This information can be used to inform people who are at greater risk of hip fracture in advance and to take more effective protective measures to reduce the incidence of hip fracture. The practical implications of the MR study are to identify those at high risk and target these groups for preventive measures.

Point 3. As described in the manuscript, obstructive sleep apnoea (OSA) is a potential confounder as it can lead to poor sleep as well as falls (which may result in hip fracture). Pain related to metastasized cancer is another potential confounder as it can disturb sleep and also affect bone health (in various ways). Neurological conditions that are associated with pain could also affect sleep as well as the likelihood of falls. Please mention confounders other than OSA.

Response 3: We thank the Reviewer for this good comment. In our observational study, we found individuals with short sleep duration and poor sleep quality were associated with higher risk of fracture risk after adjusting confounding factors. However, some unmeasured confounders might affect the relationship between sleep patterns and hip fracture. For example, bone metabolism disorder and sleep disturbance were usually simultaneously observed in patients with bone metastasis. Following the reviewer’s advice, we have added pain and mental disorders as potential confounders of sleep patterns and hip fracture in the Discussion section (Please refer lines 333-335, page 7).

Point 4. Please provide power calculations for this study. Although the sample size is seemingly large, the outcome is relatively rare.

Response 4: We thank the Reviewer for this good comment. The reason for the outcome is relatively rare despite a seemingly large sample size may be due to the incidence of hip fracture (about 2%, 224/10929) is a rare event. In this case, the power of the study could be low even with a large sample size because the probability of observing the event is small. To calculate the power for this study investigating a rare event, we used a sample size calculator “pwr” function of R package. According to the incidence of hip fracture (2%), the significance level (P=0.05), and the incidence difference between groups (0.5%), we used sample size (n=10929) could achieve the 0.70 power. Thus, when investigating a rare event, a large sample size may be required to achieve adequate power, even though the outcome is rare.

Point 5. The results as they are currently presented are somewhat confusing. Which results are based on instrumental variable modelling and which are not? Please also give a clear, structured overview of how the various analyses produced various (sometimes seemingly contradictory) results, and please provide context and interpretation for these findings.

Response 5: Thank the Reviewer for this comment. This study was divided into two parts: observational and Mendelian Randomization(MR) study.

Firstly, we examined the association of 3 sleep characteristics (sleep duration, midday napping, and sleep quality) and hip fracture in a large Chinese middle-aged and elderly cohort. Our results revealed that participants who self-reported short sleep duration and poor sleep quality (e.g., restless sleep) were associated with higher risk of fracture. However, baseline midday napping time had no effect on the risk of hip fracture outcome.

Then, MR analysis were used to infer the causal relationship of 4 sleep characteristics (sleep duration, daytime napping, chronotype, and insomnia) and hip fracture. Our results only showed that a modest causal roles of sleep duration on hip fracture. However, no evidence of a causal relationship between other sleep traits, such as chronotype, daytime napping and insomnia, and fracture was found. Above MR results were obtained by using genetic variants as instrumental variables.

The sleep characteristics examined in observational and MR study were not completely overlapped. However, both observational and MR analysis showed that short sleep duration has detrimental effect on hip fracture incidence, while no relationship between daytime napping and hip fracture was observed.

To avoid misunderstanding, we revised our results with structured headings, including observational analysis and two sample MR analysis. Observational analysis was reported with 4 paragraphs: Characteristics of the study population, self-reported sleep duration and hip fracture, self-reported midday napping and hip fracture, and self-reported sleep quality and hip fracture.

Minor concerns

Point 1.  Abstract: “…genetically predicted self-reported sleep duration …” I can understand what is meant by this after reading the full manuscript, but on first reading the abstract only, I was confused. Can you please rephrase so that the abstract can be read as stand-alone.

Response 1: Thank the Reviewer for this comment. The sentence was rephrased with following: “We further inferred the causal relationship between self-reported sleep behaviors and hip fracture using MR approach. Among 4 sleep phenotypic studies (sleep duration, daytime napping, chronotype, and insomnia), we only found a modest causal relationship between sleep duration and fracture (OR=0.69, 95%CI: 0.48 to 0.99, P=0.04). However, no causal relationship was observed for other sleep traits.” (Please refer lines 22-26, page 1).

Point 2.  Introduction, lines 33-34: please list the time and place for this statistic.

Response 2: Thank the Reviewer for this comment. In 2000, the estimated number of new osteoporotic fractures worldwide was 9.0 million, of which 1.6 million (18.2%) were at the hip[6]. By 2050, the estimated number of new hip fractures worldwide will increase to 6.26 million cases[7]. We have included above information in lines 34-36.

Point 3.  In the methods section, please provide more information on each of the surveys that were used (three surveys; one baseline and two follow-ups, at two-year intervals each?) and the participation numbers for each.

Response 3: Thank the Reviewer for this comment. China Health and Retirement Longitudinal Study (CHARLS) is a large, prospective, national cohort. The baseline population was recruited in 2011 (wave 1) from 150 counties/districts in 28 provinces and followed up every 2 years. In this study, we utilized data from the CHARLS 2011 (wave 1) and 2015 (wave 3). The baseline survey of CHARLS 2011 (wave 1) included a total of 17,708 participants, and CHARLS 2015 (wave 3) included 20,517 participants. Follow the Reviewer’s advice, we have provided above information in Methods section. (Please refer lines 99-106, page 3)

Point 4. After-lunch napping is mentioned specifically (methods, line 112). What about late-morning or early evening naps?

Response 4: Thank the Reviewer for this comment. After-lunch napping, a culturally perceived health behavior, is a common practice in China[8, 9]. Thus, midday napping data, specifically for after-lunch napping, were collected through self-reported questionnaire instead of late-morning or early evening naps.

Point 5. Did you test for an interaction effect between sleep duration and napping? If so, can you describe in the methods exactly how this was done?

Response 5: We thank the reviewer for raising this interesting issue. Given short nighttime sleep duration may increase midday napping time, we carried out joint effects analysis of sleep duration and midday napping on hip fracture incidence. We first defined the categories, though combining six categories of sleep duration with five categories of nap duration. A total of 30 more detailed categories, each of which was defined as "yes" if both were included. Then, logistic regression was carried out on each of the 30 categories and the outcome to obtain the relationship with the outcome. (Please refer lines 177-180, Page 5).

Point 6. If this study is modelled on previous studies that have the utilized genetic variants as instrumental variables in epidemiological, observational research, please refer to these.

Response 6: Thank the Reviewer for this comment. We have referred two previous studies in the revised version as followed (Please refer lines 280-282, Page 6):

  • Qian Y, Xia JW, Liu KQ, Xu L, Xie SY, Chen GB, Cong PK, Khederzadeh S, Zheng HF: Observational and genetic evidence highlight the association of human sleep behaviors with the incidence of fracture. Commun Biol 2021, 4(1).
  • Chen J, Zhang J, So HC, Ai S, Wang N, Tan X, Wing YK: Association of Sleep Traits and Heel Bone Mineral Density: Observational and Mendelian Randomization Studies. J Bone Miner Res 2021, 36(11):2184-2192

Point 7. Methods page 5 line 186 “… increased risk of hypertension”. Please use ‘prevalence’ instead of ‘risk’.

Response 7: We thank the Reviewer for this good suggestion. We have corrected it (Please refer lines 196, Page 5).

Point 8. The U curve is shown in Figure 1 but I can’t see the ‘U’ pattern in the results presented in Table 2, in the adjusted or unadjusted results. Please provide a clear explanation for this.

Response 8: We thank the Reviewer for raising this issue. Restricted cubic spline (RCS) and Logistic regression are two different statistical analysis methods, so they may yield different results when analyzing the same data. RCS is a non-parametric method that can capture complex non-linear relationships, which assumes that the relationship between the variables is continuous, so it can adapt well to non-linear relationships. In particular, when drawing curve relationships with restricted cubic splines, it is usually necessary to manually set the number (k) and position (ti) of spline function nodes. In most cases, the position of nodes has little effect on the fitting of the restricted cubic spline, while the number of nodes determines the shape of the curve, or the degree of smoothness, which may have a certain impact.

Logistic regression, on the other hand, typically assumes that the relationship between the variables is linear or approximately linear. When we conducted the logistic grouping, only 8% of the middle-aged and elderly people had more than 9 hours of sleep duration, and less than 20 of them had hip fracture. The lack of data volume itself may lead to a certain deviation between the results and the RCS curve. Finally, in the logistic regression, we set the reference group as a general sleep duration of 6-7 hours for the normal population, while in the RCS curve, we use the predicted value generated to find the sleep duration closest to "1" is 5.64 hours, which has some conflict with our artificially set reference and may also be the reason for the large gap in the data at the end of the curve. Therefore, if there is a U-shaped or inverted U-shaped relationship, RCS may be more likely to capture this relationship, while logistic regression may not be able to capture it.

Point 9. Figs 3 & 4: in the figure legends, please explain what modelling was applied to produce these results.

Response 9: We thank the Reviewer for this good suggestion. We have corrected the figure legends with following: “Figure 3 Forest plot of logistic regression analysis of sleep duration and risk of hip fracture, stratified by baseline information.” and “Figure 4 Forest plot of logistic regression analysis of Midday napping and risk of hip fracture stratified by baseline information.”

Point 10.  At the end of the manuscript (Page 8, line 369) there is a statement: “This is an observational study with no ethical approval”. This is at odds with Page 3 Lin 104-105: “This study was performed in line with the principles of the Declaration 104 of Helsinki. This study was approved by Biomedical Ethics Review Committee of Peking 105 University. All subjects enrolled signed informed consent.” I presume the latter is the correct statement?

Response 10: We thank the Reviewer for raising this issue. Indeed, the latter is the correct statement and we have corrected this mistake in Page 8, line 381-382

References

  1. Lamb SE, Bruce J, Hossain A, Ji C, Longo R, Lall R, Bojke C, Hulme C, Withers E, Finnegan S et al: Screening and Intervention to Prevent Falls and Fractures in Older People. N Engl J Med 2020, 383(19):1848-1859.
  2. Zhu XN, Chen L, Pan L, Zeng YX, Fu Q, Liu YB, Peng YD, Wang YF, You L: Risk factors of primary and recurrent fractures in postmenopausal osteoporotic Chinese patients: A retrospective analysis study. Bmc Womens Health 2022, 22(1).
  3. Dewan N, MacDermid JC, Grewal R, Beattie K: Risk factors predicting subsequent falls and osteoporotic fractures at 4 years after distal radius fracture-a prospective cohort study. Arch Osteoporos 2018, 13(1):32.
  4. Slemenda C: Prevention of hip fractures: risk factor modification. Am J Med 1997, 103(2A):65S-71S; discussion 71S-73S.
  5. Bowden J, Holmes MV: Meta-analysis and Mendelian randomization: A review. Research synthesis methods 2019, 10(4):486-496.
  6. Johnell O, Kanis JA: An estimate of the worldwide prevalence and disability associated with osteoporotic fractures. Osteoporos Int 2006, 17(12):1726-1733.
  7. Veronese N, Maggi S: Epidemiology and social costs of hip fracture. Injury 2018, 49(8):1458-1460.
  8. Zhou L, Yu K, Yang L, Wang H, Xiao Y, Qiu G, Liu X, Yuan Y, Bai Y, Li X et al: Sleep duration, midday napping, and sleep quality and incident stroke: The Dongfeng-Tongji cohort. Neurology 2020, 94(4):e345-e356.
  9. Mo T, Wang Y, Gao H, Li W, Zhou L, Yuan Y, Zhang X, He M, Guo H, Long P et al: Sleep Duration, Midday Napping, and Serum Homocysteine Levels: A Gene-Environment Interaction Study. Nutrients 2023, 15(1).

For more details please see the revised version manuscript.

Reviewer 2 Report

The paper il quite interesting and it is a good point of view for new approach in personalized - precision medicine.

Major

The relationship between endogenic factor and fractures is interesting. But I think it is more correct think about an association between risk of fall and sleep. Or to motivate because authors assess the relationship directly with fractures, it seems an association with specific endogenic factors. There are some samples of genetic factors associated to pathologies (RANK/RANKL/OPG pathway: genetic association with history of ischemic stroke, Biscetti et al). Authors need to better specify.

A big concern:

The study design is not clear, the association is 'with fractures' or 'with risk of falls'? 

Author Response

Dear Reviewer,

Response to Reviewer 2 Comments

The paper il quite interesting and it is a good point of view for new approach in personalized - precision medicine.

Major

Point 1. The relationship between endogenic factor and fractures is interesting. But I think it is more correct think about an association between risk of fall and sleep. Or to motivate because authors assess the relationship directly with fractures, it seems an association with specific endogenic factors. There are some samples of genetic factors associated to pathologies (RANK/RANKL/OPG pathway: genetic association with history of ischemic stroke, Biscetti et al). Authors need to better specify.

Response 1: We thank the Reviewer for this good suggestion. Genetic variation is stable across the human lifespan and identification of genetic factors for disease may help optimizing effective prevention approaches. Sleep disturbances are common in older population and may influence bone health, falls, and fracture directly by influencing bone turnover and muscle strength or indirectly through high comorbidity or poor physical function. In the present study, we applied genetic variants as instrumental variables to assess the causality of sleep-fracture association. Recently, a large-scale genome-wide association analysis on falling risk in 89,076 cases and 362,103 controls from the UK Biobank Study have found risk for falling has a strong positive genetic correlation with fracture[1]. The analysis revealed a small, but significant SNP-based heritability (2.7%). As more fall-associated loci were identified, it will allows us to investigate causal relationships between sleep traits and falls in future.

A big concern:

Point 2. The study design is not clear, the association is 'with fractures' or 'with risk of falls'? 

Response 2: We thank the Reviewer for this good suggestion. The goal of this study was mainly to examine the effects of sleep behaviors on hip fracture within a large middle-aged and older cohort and then infer their causal role with Mendelian randomization (MR) analysis. Falls are the most common risk factor of fractures among older population, thus many studies will include falling when fracture is mentioned, especially in those with osteoporosis[2-5]. Given falls account for high proportion of fractures in the elderly[6, 7], this study defines the occurrence of hip fracture as an outcome instead of the risk of falling.

References

  1. Trajanoska K, Seppala LJ, Medina-Gomez C, Hsu YH, Zhou SR, van Schoor NM, de Groot LCPGM, Karasik D, Richards JB, Kiel DP et al: Genetic basis of falling risk susceptibility in the UK Biobank Study. Commun Biol 2020, 3(1).
  2. Karinkanta S, Piirtola M, Sievanen H, Uusi-Rasi K, Kannus P: Physical therapy approaches to reduce fall and fracture risk among older adults. Nat Rev Endocrinol 2010, 6(7):396-407.
  3. Barker AL, Morello R, Thao LTP, Seeman E, Ward SA, Sanders KM, Cumming RG, Pasco JA, Ebeling PR, Woods RL et al: Daily Low-Dose Aspirin and Risk of Serious Falls and Fractures in Healthy Older People: A Substudy of the ASPREE Randomized Clinical Trial. JAMA Intern Med 2022, 182(12):1289-1297.
  4. Lamb SE, Bruce J, Hossain A, Ji C, Longo R, Lall R, Bojke C, Hulme C, Withers E, Finnegan S et al: Screening and Intervention to Prevent Falls and Fractures in Older People. N Engl J Med 2020, 383(19):1848-1859.
  5. Pan F, Tian J, Cicuttini F, Jones G: Sleep disturbance and bone mineral density, risk of falls and fracture: Results from a 10.7-year prospective cohort study. Bone 2021, 147.
  6. Ambrose AF, Cruz L, Paul G: Falls and Fractures: A systematic approach to screening and prevention. Maturitas 2015, 82(1):85-93.
  7. Vranken L, Wyers CE, Van der Velde RY, Janzing HMJ, Kaarsemakers S, Driessen J, Eisman J, Center JR, Nguyen TV, Tran T et al: Association between incident falls and subsequent fractures in patients attending the fracture liaison service after an index fracture: a 3-year prospective observational cohort study. BMJ Open 2022, 12(7):e058983.

For more details please see the revised version manuscript.

Round 2

Reviewer 1 Report

Thank you for the thorough response and revision.

Can you please include your response to points 2 and 4 in the revised manuscript?

Author Response

Thank you for the thorough response and revision.

Point 1. Can you please include your response to points 2 and 4 in the revised manuscript?

Response 1: Thank the Reviewer for this good suggestion. We have included the response to points 2 and 4 in the revised manuscript.

Reviewer 2 Report

The paper is suitable for publication

Author Response

The paper is suitable for publication

Response: We are truly appreciative of the reviewer’s constructive comments.